# Hypothalamic NAD^+^-Sirtuin Axis: Function and Regulation

**DOI:** 10.3390/biom10030396

**Published:** 2020-03-04

**Authors:** Eun Roh, Min-Seon Kim

**Affiliations:** 1Division of Endocrinology and Metabolism, Department of Internal Medicine, Korea University Guro Hospital, Seoul 08308, Korea; roheun@gmail.com; 2Division of Endocrinology and Metabolism, Department of Internal Medicine, University of Ulsan College Medicine, Asan Medical Center, Seoul 05505, Korea

**Keywords:** NAD^+^, hypothalamus, sirtuins, obesity, aging, energy metabolism, circadian rhythm

## Abstract

The rapidly expanding elderly population and obesity endemic have become part of continuing global health care problems. The hypothalamus is a critical center for the homeostatic regulation of energy and glucose metabolism, circadian rhythm, and aging-related physiology. Nicotinamide adenine dinucleotide (NAD^+^)-dependent deacetylase sirtuins are referred to as master metabolic regulators that link the cellular energy status to adaptive transcriptional responses. Mounting evidence now indicates that hypothalamic sirtuins are essential for adequate hypothalamic neuronal functions. Owing to the NAD^+^-dependence of sirtuin activity, adequate hypothalamic NAD^+^ contents are pivotal for maintaining energy homeostasis and circadian physiology. Here, we comprehensively review the regulatory roles of the hypothalamic neuronal NAD^+^-sirtuin axis in a normal physiological context and their changes in obesity and the aging process. We also discuss the therapeutic potential of NAD^+^ biology-targeting drugs in aging/obesity-related metabolic and circadian disorders.

## 1. Introduction

There has been a renaissance in nicotinamide adenine dinucleotide (NAD^+^) biology research over the past two decades since its novel functions as a signaling molecule have been revealed [1], in addition to its well-established role in cellular redox control. Substantial evidence now supports the importance of NAD^+^ biosynthetic pathways in normal metabolic regulation and the development of metabolic diseases. The hypothalamus, which is a small brain region, has long been described as a critical center that governs various metabolic processes and the circadian physiology [2]. Recently, it has also been suggested as an important area for dictating systemic aging processes [3]. This review summarizes the findings of prior studies on hypothalamic NAD^+^ biology in relation to health and disease. In particular, we focus on the involvement of NAD^+^-dependent enzymes—the sirtuins—in the onset of aging—and obesity—related metabolic complications. We also review preventive and therapeutic trials that have tested metabolic intermediates of NAD^+^ biological pathways.

## 2. NAD Biosynthetic Pathways

NAD^+^ can be synthesized de novo from the amino acid tryptophan through multiple enzymatic steps which produce nicotinic acid mononucleotide (NaMN) and then nicotinic acid dinucleotide (NaAD) by the NMN/NaMN adenylyl transferases (NMNATs) [4]. Another important NAD^+^-biosynthetic pathway is the so-called salvage pathway, via which most cellular NAD^+^ is synthesized in mammals. This pathway reuses nicotinamide or nicotinic acid produced by enzyme reactions that degrade NAD^+^. In the NAD^+^ salvage pathway of mammals, nicotinamide is directly converted to nicotinamide mononucleotide (NMN) by nicotinamide phosphoribosyl transferase (NAMPT)—the rate-limiting enzyme in this process [5,6]—and thus NAMPT is considered to be a critical regulator of cellular NAD^+^ levels. NMN is then converted to NAD^+^ by NMNATs. NMNAT1 is distributed in the nucleus, NMNAT2 in the cytosol, and NMNAT3 in the mitochondria. NAD^+^ is synthesized by the different subtypes of NMNATs in a cellular compartment-specific manner [7]. NAD^+^ can also be synthesized by the conversion of nicotinamide riboside (NR)—a trace nutrient in foods [8]. NR is converted to NMN by NR kinase and then to NAD^+^ by NMNAT.

## 3. NAD^+^-Consuming Pathways

NAD^+^ is reduced to NADH during cellular redox reactions or degraded to nicotinamide or nicotinic acid in cellular processes catalyzed by three types of NAD^+^-consuming enzymes: 1) sirtuins, 2) poly(ADP-ribose) polymerases (PARPs), and 3) cyclic ADP-ribose cyclase/cluster of differentiation 38 (CD38) [9,10,11]. The activities of these enzymes are dependent on the cellular NAD^+^ levels as NAD^+^ is a substrate (CD38) or co-substrate (sirtuins, PARPs) for the reactions of these enzymes. Hence, a disruption of normal NAD^+^ biosynthesis can affect the activities of NAD^+^-consuming enzymes, leading to altered cellular functions [12,13,14]. The overall cellular NAD^+^ contents are controlled by a balance between the activities of NAD^+^-synthesizing and -consuming enzymes [15,16,17,18].

Sirtuins are evolutionally conserved, NAD^+^-dependent deacetylases that play a broad range of roles in the regulation of transcription, energy metabolism, cell survival, DNA repair, inflammation, and the circadian rhythm [19,20]. These enzymes use one NAD^+^ molecule and generate acetyl-ADP-ribose and nicotinamide during the deacetylation reaction [21]. Among the seven mammalian sirtuins (SIRT1–7), SIRT1–3 and 7 have deacetylase activity, but SIRT4–6 have very weak or no detectable deacetylase activity [22]. SIRT6 effectively deacylates long-chain fatty acids, whilst it has little deacetylase activity [23]. Some sirtuins also exhibit ADP ribosyltransferase, demalonylation, and desuccinylation activities [24,25,26] and all of these enzymatic activities specifically require NAD^+^.

SIRT1 is the most extensively studied of the sirtuins and is mainly found in the nucleus and cytosol. SIRT1 shares the highest sequence homology with the yeast Sir2 enzyme [27], which mediates caloric restriction-induced longevity [28]. SIRT1 regulates various metabolic processes in response to changes in NAD^+^ availability and also controls cellular adaptive responses to stresses in different organs and tissues [19]. These SIRT1 actions are largely achieved via the deacetylation of key transcriptional regulators, such as peroxisome proliferator-activated receptor-γ (PPARγ), PPARγ coactivator-1α (PGC-1α), forkhead box protein O (FOXO), and nuclear factor-kappa B (NF-κB) [29,30,31,32]. SIRT2 is another major sirtuin in the cytosol and its regulatory roles have been demonstrated in adipogenesis, fatty acid oxidation, gluconeogenesis, inflammation, and oxidative stress [33]. In contrast, SIRT3–5 are distributed in the mitochondria and have regulatory roles in mitochondrial function and oxidative stress [34,35,36]. Both SIRT6 and SIRT7 are nuclear sirtuins involved in genomic stability and DNA repair [37,38,39,40].

PARPs critically mediate the DNA damage response [41,42,43]. Among the 17 members of the PARP superfamily, PARP1 plays a dominant role as a DNA-repair enzyme by which it promotes cell survival [44,45]. Under DNA damage-induced conditions, the PARPs consume up to 90% of cellular NAD^+^ in a very short period of time to catalyze massive levels of ADP-ribosylation at the sites of DNA lesions [46].

CD38 is a multifunctional, transmembrane enzyme ubiquitously distributed in mammalian tissues and has been implicated in the modulation of immune cell functions through the generation of the secondary signaling messenger cyclic-ADP-ribose (cADPR) [47]. Notably, however, recent studies have reported that the main enzymatic activity of CD38 is the hydrolysis of NAD^+^ to nicotinamide and ADPR [17,48]. CD38 is believed to hydrolyze almost 100 molecules of NAD^+^ to generate one molecule of the secondary messenger cADPR [49]. The tissue NAD+ levels in CD38-deficient mice are 10- to 20-fold higher than in wild-type animals [17]. Consistently, NADase activity is almost absent in several tissues from CD38-deficient mice [17]. These data support the novel concept that CD38 may be a major consumer of cellular NAD^+^.

## 4. Hypothalamic Regulation of Energy Homeostasis and the Circadian Rhythm

The hypothalamus is the region of the brain that is primarily responsible for the regulation of energy homeostasis and body weight [2,50,51]. The hypothalamic neurons continuously monitor the body’s energy state and modulate feeding behavior and energy expenditure (EE) to achieve energy homeostasis. The arcuate nucleus (ARC) of the hypothalamus is anatomically adjacent to the median eminence—one of the blood–brain barrier-defective circumventricular organs. Therefore, the ARC is a hypothalamic structure that first senses the peripheral metabolic signals delivered via the systemic circulation [52]. Two groups of neurons in the ARC play a crucial role in regulating energy balance and body weight: one producing proopiomelanocortin (POMC), which is a precursor of anorexigenic α-melanocyte stimulating hormone (α-MSH), and the other producing orexigenic neuropeptide Y (NPY) and *Agouti*-related peptide (AGRP). The α-MSH exerts anorexigenic and EE-promoting actions through the melanocortin-3 and -4 receptors (MC3R and MC4R) on the second-order neurons that receive axonal projections from POMC neurons [52,53]. AGRP competes with α-MSH to bind MC3R and MC4R, and thus antagonizes α-MSH actions [54,55]. However, AGRP can also act independently of α-MSH [56]. Hypothalamic POMC and AGRP neurons and their receptors are designated as the hypothalamic melanocortin system. Notably, POMC and AGRP neurons are oppositely regulated by circulating metabolic signals. For instance, the adipocyte-derived hormone leptin stimulates POMC neuronal activity, but inhibits that of AGRP [53,57,58]. The ventromedial nucleus of the hypothalamus (VMH) has also been recognized as an important structure for determining the energy balance [59]. Steroidogenic factor-1 (SF1)-positive neurons in the VMH respond to changes in circulating leptin and other metabolic cues [60] and thereby regulate energy homeostasis [61,62].

In mammals, the principal circadian pacemaker neurons reside in the hypothalamic suprachiasmatic nucleus (SCN) [63,64]. The SCN is described as a “master synchronizer”, as the SCN clock neurons synchronize subordinate cellular clocks across the body to a uniform internal time [65]. The SCN also coordinates the circadian control of behavioral (for example, feeding–fasting and sleep–wakefulness), neuroendocrine, and autonomic nervous systems through the entrainment of cellular clocks in target tissues [66,67]. At the molecular level, the cellular clock is composed of a transcription-translation feedback loop. The heterodimeric complexes of the transcription factors circadian locomotor output cycle protein kaput (CLOCK) and brain and muscle ARNT-like 1 (BMAL1) stimulate the expression of clock genes such as period (PER)-1, -2, and -3 and cryptochrome (CRY)-1 and -2. These clock genes in turn suppress CLOCK/BMAL1 transcriptional activity and operate in a negative feedback loop by repressing their own transcription [68,69,70,71].

## 5. Regulatory Roles of the Hypothalamic NAD^+^-Sirtuin Axis in Normal Physiological Conditions

The role of hypothalamic sirtuins, specifically SIRT1, in the control of body weight and feeding behavior has been extensively studied. SIRT1 is expressed in most brain regions, most prominently in the hypothalamus [72,73]. Hypothalamic SIRT1 mediates the metabolic adaptation to altered energy conditions [74,75,76,77,78,79]. Fasting increases hypothalamic SIRT1 protein expression and NAD^+^ contents, leading to increased SIRT1 deacetylase activity [74]. The pharmacological inhibition or knockdown of hypothalamic SIRT1 suppresses food intake and body weight gain in rodents on a regular chow diet (RCD) [74]. As for the underlying molecular mechanism, SIRT1 deacetylates FOXO1 and potentiates the FOXO1-mediated regulation of POMC and AGRP transcription. These results suggest that SIRT1 functions as a critical regulator of the hypothalamic melanocortin system.

Subsequent studies have indicated that the metabolic effects of SIRT1 may be cell type-specific and metabolic context-dependent. SIRT1 deletion specifically in POMC neurons does not significantly alter body weight or adiposity in the RCD-fed condition. In contrast, these mutant mice on a high-fat diet (HFD) are vulnerable to diet-induced obesity due to reduced EE, because SIRT1 in POMC neurons is required for the brown adipose tissue-like remodeling of the perigonadal white adipose tissue through sympathetic activation [76]. Likewise, mice lacking SIRT1 in VMH SF1 neurons are more susceptible to HFD-induced obesity and type 2 diabetes mellitus, whereas the mutant mice on an RCD display a normal body weight [77]. Interestingly, SIRT1 in SF1 neurons controls insulin sensitivity in skeletal muscle [77]. These data indicate that SIRT1 is essential for the normal functions of POMC and SF1 neurons that provide resistance to weight gain and the development of metabolic complications upon exposure to a HFD. SIRT1 in NPY/AGRP neurons also appears to be essential for their functions. The pharmacological inhibition or knockout of SIRT1 in AGRP neurons decreases the responses to the orexigenic gut hormone ghrelin. Consequently, these manipulations decrease food intake and lower fat mass and body weight [78,80]. Hence, diminished SIRT1 activity in AGRP neurons may be accountable for the anorexigenic effect induced by hypothalamic SIRT1 knockdown [74]. According to a recent study [81], the hypothalamic expression levels of the NAD^+^ biosynthetic enzyme NAMPT are affected by the administration of metabolic hormones (ghrelin and leptin) or HFD consumption. Furthermore, hypothalamic NAMPT inhibition with its chemical inhibitor FK866 abrogates nocturnal feeding and ghrelin-induced hyperphagia. These data suggest that the NAD^+^ salvage pathway in hypothalamic neurons may be involved in the adaptive regulation of feeding behavior to metabolic alterations.

Previous studies have demonstrated that the biology of intracellular NAD^+^ is closely linked to the cellular clock [82,83,84,85]. SIRT1 has a pivotal role in the circadian oscillation of molecular clock gene expression. It binds the CLOCK-BMAL1 complex in a circadian manner to inhibit its transcriptional activity [83,84]. On the other hand, SIRT1 has been shown to promote PER2 degradation through deacetylation [85]. A subsequent study reported that SIRT1 directly stimulates BMAL1 transcription via PGC-1α, resulting in an increased amplitude in the circadian oscillation of SCN circadian gene expression [86]. Brain-specific SIRT1 knockout mice exhibit reduced circadian gene expression in the anterior hypothalamus where the SCN is located, suggesting that SIRT1 upregulates the transcription of clock genes [86]. Interestingly, the SCN expression of SIRT1 and NAMPT were found to oscillate in a circadian manner [86]. Another study has reported that SIRT1 in VMH SF-1 neurons mediates the relay of nutritional input information to the SCN central clock to synchronize it to feeding cues [87]. Since cellular and tissue NAD^+^ contents are a major determinant of SIRT1 deacetylase activity, the findings of these prior studies strongly indicate that the NAMPT-SIRT1 axis is a nutrient sensor that couples body metabolism to the circadian rhythm of the central clock [85,87].

Further evidence suggests that SIRT1 may regulate mammalian aging and longevity through hypothalamic mechanisms. Brain-specific SIRT1-overexpressing mice display enhanced neuronal activity in the DMH and lateral hypothalamus (LH) and resistance to the aging-associated decline in skeletal muscle mitochondrial function. They also exhibit improved physical activity, thermogenesis, and oxygen consumption, as well as better sleep quality upon aging [88]. The molecular basis of SIRT actions in regulating DMN and LH neural activation involves its interaction with Nk2 homeobox 1 (NKX2-1) and subsequent upregulation of orexin type 2 receptor (OX2R) expression [88]. Indeed, a DMH/LH-specific SIRT1 knockdown study has further demonstrated the importance of the SIRT1-NKX2-1-OX2R-mediated signaling pathway in counteracting age-associated physiological decline. These data indicate that SIRT1 and its partner NKX2-1 in the hypothalamic DMH and LH play an indispensable role in retarding aging and promoting longevity [88].

Several studies have reported that hypothalamic SIRT1 modulates neuroendocrine pathways by affecting hormone synthesis and secretion in the hypothalamus and pituitary gland. Hypothalamic SIRT1 has been reported to control the thyroid function through complex formation with FOXO1 and Necdin [89]. SIRT1 and Necdin promote FOXO1 deacetylation, leading to the activation of the hypothalamic-pituitary-thyroid axis [89]. Other lines of evidence have documented a regulatory role of SIRT1 in the hypothalamic-pituitary-gonadal axis [90,91,92]. Mice with an SIRT1 deletion display defective spermatogenesis at the postnatal stage due to reduced hypothalamic GnRH expression [91]. However, detailed mechanisms of GnRH regulation by SIRT1 remain to be fully elucidated. On the other hand, the depletion of SIRT6 in a neuron-specific manner lowered plasma growth hormone (GH)/insulin-like growth factor-1 (IGF-1) levels and decreased hypothalamic POMC expression, which led to postnatal growth retardation and obesity [93]. Therefore, neuronal SIRT6 may critically regulate somatic growth and energy balance through chromatin remodeling and the transcriptional regulation of related genes.

## 6. Disrupted Hypothalamic NAD^+^ Biology in Obesity and Aging 

Obesity is a condition caused by long-term positive imbalances in energy intake and expenditure [94]. Over the past 50 years, obesity has become the fastest-growing global health problem, with a significant economic burden [95,96]. The global emergence of obesity coincidences with the global aging phenomenon [96,97]. The chronic consumption of a caloric-rich diet impairs the ability of hypothalamic neurons to sense metabolic signals and maintain energy and glucose homeostasis [98,99,100]. As mentioned in the previous section, NAD^+^-dependent SIRT1 activity is pivotal for the hypothalamic regulation of energy homeostasis and circadian physiology [74,76,77,78,79,86], which are disrupted during the process of obesity and aging [101,102]. Hence, several studies have investigated alterations in the metabolic organ NAD^+^ levels and SIRT1 activity in rodent models of obesity and aging. These reports have demonstrated that NAMPT-mediated NAD^+^ biosynthesis is severely compromised in multiple metabolic organs in mice on an HFD, along with reduced tissue NAD^+^ levels [103]. Specifically, the hypothalamic NAD^+^ content is significantly reduced in diet-induced obese mice [81,104,105] and genetically-obese db/db mice [104]. Similarly, the NAMPT expression levels and NAD^+^ bioavailability decline in the brain and other metabolic tissues of aged rodents and humans [13,81,103,106,107,108]. Therefore, reduced NAD^+^ contents in hypothalamic nuclei, including ARC, VMH, and SCN, may account for hypothalamic dysfunction observed in obesity and aging: altered energy metabolism, insulin resistance, and circadian rhythm disruption (Figure 1). However, conflicting results have been reported in regard to hypercaloric diet- or aging-induced changes in metabolic organ NAMPT and NAD^+^ levels. Hepatic NAD^+^ contents and NAMPT expression were shown to be elevated in mice on an HFD and suggested as a compensatory mechanism to protect against hepatic lipid accumulation [109]. In another study, hepatic NAD^+^ and NAMPT levels were not altered by the consumption of an HFD [110]. These discrepant results from animal studies might be caused by the HFD composition and consumption duration, as well as by the rodent strains used. Conflicting human data on the metabolic organ NAMPT expression have also been reported. Obese women with hepatic steatosis showed higher hepatic NAMPT mRNA expression compared to lean women [111]. In a recent paper, aged humans exhibited reduced NAMPT protein levels in skeletal muscle, but no change in adipose tissue, compared with those of young subjects [112].

The PARPs have been suggested to be responsible for the aging-related NAD decline in metabolic tissues [11,15,16]. Indeed, PARP activity, measured by the global poly-ADP-ribosylation of cellular proteins, is chronically activated in aged worms and mice, which leads to a reduction in the NAD^+^ content and SIRT1 activity levels [113]. Indeed, aging is associated with increased nuclear DNA damage, which may result in increased PARP activity and NAD^+^ depletion [113]. Consistently, mice lacking PARP1 show increased tissue NAD^+^ levels and enhanced SIRT1 activity [15]. In contrast, it was reported that neither PARP1 nor NAMPT expression in human adipose tissues were altered with aging [114]. Instead, the authors found that the CD38 expression levels were significantly elevated during the chronological aging process, implying a critical contribution of CD38 in aging-related tissue NAD^+^ deficiency. They further found that enhanced CD38 expression caused aging-related mitochondrial dysfunction via mitochondrial SIRT3-dependent mechanisms [114]. Further investigations will be needed to address whether CD38 may also be involved in the aging-induced decline in hypothalamic NAD^+^ contents [115].

In addition to reduced NAD^+^ contents, hypothalamic SIRT1 protein expression is decreased by the consumption of HFD or a high-sucrose diet [104], and this change may also contribute to decreased hypothalamic SIRT1 activity. Therefore, the consumption of a hypercaloric diet mitigates the beneficial effects of ARC SIRT1 overexpression by decreasing both SIRT1 protein expression and hypothalamic NAD^+^ contents [105]. Similar to diet-induced obesity, hypothalamic SIRT1 protein expression levels also decrease with age [104], which may contribute to age-associated weight gain. The aging-related decrease in hypothalamic SIRT1 activity causes a dampened expression of the clock components in the hypothalamic SCN and this phenomenon has been suggested as a central mechanism underlying aging-related impairment in the circadian function [87].

Microglia are brain-resident, principal innate immune cells that engage various neuroinflammatory processes [116]. Obesity and aging accompany the activation of microglia and inflammatory signaling pathways in the hypothalamus [117,118]. Hypothalamic inflammation is now being regarded as a key mechanism underlying hypothalamic neuronal dysfunction in obesity and aging [3,119]. Notably, SIRT2 represses microglia-mediated inflammation and neurotoxicity through NF-κB deacetylation and inhibition [120]. An obesity- and aging-associated decline in NAD^+^ content and SIRT2 activity in hypothalamic microglia might exaggerate microglial activation and hypothalamic inflammation and accelerate systemic aging and obesity.

## 7. Therapeutic Trials Targeting Hypothalamic NAD^+^ Biology 

Since NAD^+^-dependent SIRT1 activity is essential for a normal metabolic function [20], treatments that target the cellular NAD^+^-SIRT1 axis represent a potential preventive/therapeutic intervention strategy for obesity- and aging-related metabolic disorders (Figure 2). Previous studies have demonstrated the metabolic benefits of SIRT1-activating compounds, i.e., natural SIRT1 activators (resveratrol) and synthetic small-molecule SIRT1 activators [121,122,123,124,125,126]. In mice and humans, resveratrol has been found to improve HFD-induced metabolic complications [121,124], as well as aging-related neurodegenerative diseases, such as Alzheimer’s disease, Parkinson’s disease, and stroke [122,126]. In contrast, resveratrol treatment did not show significantly beneficial effects in normal lean rodents and non-obese human subjects [127,128]. Small SIRT1 activators that act via an allosteric mechanism have been developed and these synthetic compounds are more potent than naturally occurring compounds [129]. These new compounds can reverse caloric-rich diet-induced metabolic alterations [129] and extend the life span in mice, even when they are fed a standard diet [130]. Notably, however, therapeutic strategies that use these compounds may have a limitation because they can only increase SIRT1 activity under conditions of an adequate cellular NAD^+^ content.

Supplementation with the NAD^+^ precursors NMN and NR has been used in therapeutic trials for age- and hypercaloric diet-induced pathophysiologies [13,103,131,132]. The intraperitoneal administration of NMN was shown to significantly increase the NAD^+^ contents and SIRT1 activity in metabolic organs such as the liver, pancreas, and white adipose tissue and to improve the glucose metabolism in diabetic mice [103]. Moreover, increasing the NAD^+^ levels through NMN administration in aged mice restored the mitochondrial function of skeletal muscles to the levels of young mice via SIRT1-dependent mechanisms [13]. In line with these findings, supplementation with NR in HFD-fed mice efficiently increased the NAD^+^ contents in multiple metabolic tissues and activated both SIRT1 and SIRT3 activity. Furthermore, NR supplementation curbed the detrimental metabolic effects caused by HFD feeding [131]. A recent study also demonstrated that a 12 month administration of NMN in mice attenuated their age-associated physiological decline, including body weight gain, without any obvious toxicity [132].

These excellent outcomes of NAD^+^ precursor supplementation in rodents prompted the initiation of human clinical studies [133,134,135,136]. The phase I human clinical trial has been launched, in order to study the beneficial effects of NMN supplementation in humans [133,134]. Moreover, six weeks of oral NR supplementation in healthy middle-aged or elderly human subjects effectively increased NAD^+^ metabolism in circulating blood mononuclear cells, and induced beneficial vascular changes, and exhibited a trend towards a decreased blood pressure and aortic stiffness [135]. In line with this, aged people taking NR supplements for three weeks showed increased skeletal muscle NAD^+^ metabolites and decreased circulating inflammatory cytokine levels [136]. Contrary to these findings, 12 weeks of dietary NR supplementation did not alter the insulin sensitivity, whole-body glucose metabolism, β-cell insulin secretory capacity, and gut incretin hormone secretion in obese men with insulin resistance [137,138]. Moreover, NR supplementation had no significant effect on the skeletal muscle mitochondrial respiration, content, or morphology in those subjects [139]. Future studies with higher doses of NR for a longer treatment duration in both sexes may be needed to document the therapeutic benefit of NR supplementation in humans.

NR enters cells via nucleoside transporters, without conversion to other intermediates. It has been proposed that NMN is converted to NR extracellularly by CD73 ecto-5′-nucleotidase [140] and then transported into cells via nucleoside transporters. Upon entering cells, NR is then reconverted to NMN. Orally-administered NMN is rapidly absorbed from the gut into the blood circulation and transported into tissues, thereby affectively increasing the NAD^+^ content in tissues over one hour [132]. It has recently been reported that the *Slc12a8* gene encodes a specific NMN transporter expressed in the mouse small intestine [141]. In support of this putative role, a lack of *Slc12a8* leads to decreased NAD^+^ levels in the jejunum and ileum due to a reduced NMN uptake. Although it remains unclear whether NMN enters the brain across the blood–brain barrier (BBB), intraperitoneal NMN administration has been found to rapidly increase the NAD^+^ levels in brain regions such as the hippocampus and hypothalamus [142,143].

Our own research group has examined the effects of NAD supplementation on energy metabolism in mice. We found that a single administration of NAD via central and peripheral routes suppressed fasting-induced hyperphagia and weight gain in lean mice when administered in overnight-fasted mice [144]. Our findings seem to be contradictory to the findings of a recent study [82]. In that study, the intracerebroventricular administration of NAMPT inhibitor FK866 inhibited ghrelin-induced feeding and also suppressed nocturnal feeding when administered before light-off [82]. These conflicting results might be caused by the differences in experimental settings or animals. It should be noted, however, that SIRT1 in POMC and SF1 neurons acts to resist weight gain, whereas SIRT1 in NPY/AGRP neurons is required for fasting- and ghrelin-induced hyperphagia [79]. Therefore, the therapeutic elevation of hypothalamic NAD^+^ contents might generate differential outcomes, according to the metabolic state during experiments.

We also demonstrated in another report that four weeks of NAD supplementation significantly attenuated weight gain in HFD-fed mice, but not in chow diet-fed lean mice, without any detectable side effects [105]. Exogenous NAD supplementation normalized the reduced hypothalamic NAD^+^ content in diet-induced obese mice. Notably, the effective dose of NAD required to induce beneficial metabolic effects was 100-times lower than that of NMN or NR. Exogenous NAD is transported into the hypothalamus via the gap junction/hemichannels protein connexin 43, which is highly expressed in the astrocytes and tanycytes located at the blood–brain barrier and the blood–cerebrospinal fluid barrier [145,146]. In addition, chronic NAD supplementation helps obese mice to recover from a blunted diurnal rhythm in their locomotor activity and partially improved the diurnal rhythmicity in feeding behaviors [105]. Chronic NAD supplementation also improved the diminished diurnal oscillation of PER1 expression in the hypothalamic ARC, but not in the SCN. These results indicated that the hypothalamic ARC is a potential site of action for NAD supplements in terms of diurnal rhythm regulation. Our findings highlighted the therapeutic potential of NAD supplementation in obese patients with disrupted circadian rhythms.

The central nervous system has been suggested to have a relatively lower intracellular NAMPT activity (iNAMPT) compared to other organs and tissues [147]. The brain may be more vulnerable to diseases related to inadequate NAD^+^ biosynthesis [148]. In this regard, NAMPT is secreted from cells such as adipocytes and is found in the circulating blood, referred to as extracellular NAMPT (eNAMPT). eNAMPT can promote NAD^+^ biosynthesis in remote organs by providing the NAD^+^ precursor NMN to the organs with lower iNAMPT activity [143]. Supporting this hypothesis, adipose tissue-specific NAMPT knockout mice showed a reduction in circulating eNAMPT levels, as well as in the hypothalamic NAD^+^ content and SIRT1 activity [143]. Interestingly, these changes were only observed in females and the mechanism of the gender difference remains unknown. Conversely, adipose tissue-specific NAMPT knockin mice showed the opposite changes in these parameters, which were observed in both males and females. These findings provide convincing evidence that eNAMPT secreted by adipose tissue can remotely regulate hypothalamic NAD^+^ biosynthesis and SIRT1 activity. Considering these findings, eNAMPT may potentially be applied to the treatment of disorders related to disrupted hypothalamic NAD^+^ biology. However, plasma eNAMPT concentrations are elevated in humans with visceral obesity and diabetes [149,150], which may mitigate the therapeutic potential of eNAMPT supplementation.

The inhibition of CD38, which is one of the NAD^+^-consuming enzymes, has been suggested as a possible pharmacological strategy for treating aging-related tissue NAD^+^ deficits and metabolic dysfunction [114,151]. Indeed, a recent study has reported that, in aged mice, treatments with the highly potent CD38 inhibitor 78c attenuated the age-related NAD^+^ decline and promoted a significant improvement in several physiological parameters, including glucose homeostasis, the exercise capacity, muscle architecture, and cardiac function [18]. As these effects were independent of PARP activity [18], further studies will be needed to elucidate the mechanisms underlying the anti-aging effects of 78c, especially in the hypothalamus.

## 8. Concluding Remarks

Research data obtained from the genetic manipulation of enzymes involved in NAD^+^ biosynthesis or consumption have indicated that impaired NAD^+^ biology in hypothalamic neurons significantly contributes to diet- and aging-induced metabolic disorders and circadian disruption. Moreover, trials targeting hypothalamic NAD^+^ biology have revealed therapeutic benefits in terms of reversing the hypothalamic dysfunction that occurs during the progression of aging and diet-induced obesity, as well as in promoting healthy aging. It should be noted that the exogenous supplementation of the NAD^+^ precursors NMN and NR, and NAD^+^ itself, effectively recover the NAD^+^ deficiency in the hypothalamus in rodents. Future studies are warranted to reveal the metabolic pathways and transport mechanisms for exogenous NAD^+^ precursors and NAD^+^ in the hypothalamus. Moreover, the therapeutic potential of NAD^+^ biology-targeting drugs should be precisely tested in humans undergoing aging and also in obese subjects.

## Figures and Tables

**Figure 1 biomolecules-10-00396-f001:**
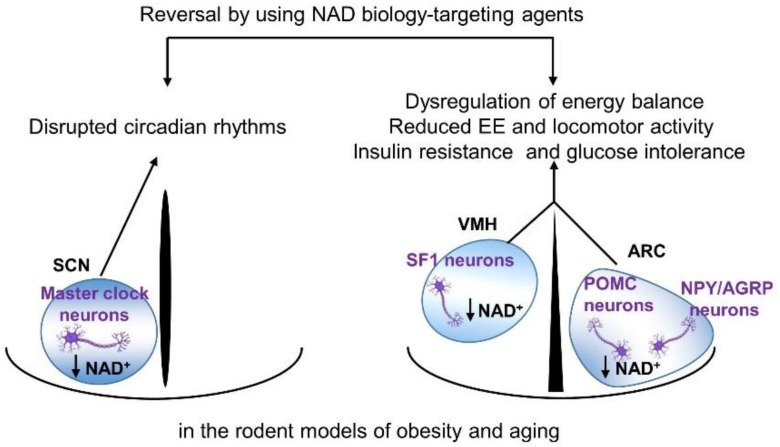
Altered NAD^+^ biology in the hypothalamic neurons leads to the disruption of energy/glucose metabolism and circadian rhythms observed in obese or aged humans and rodents. AGRP: *Agouti*-related protein, ARC: arcuate nucleus, NAD^+^: nicotinamide adenine dinucleotide, NPY: neuropeptide Y, POMC: proopiomelanocortin, SCN: suprachiasmatic nucleus, SF1: steroidogenic factor-1, and VMH: ventromedial hypothalamus.

**Figure 2 biomolecules-10-00396-f002:**
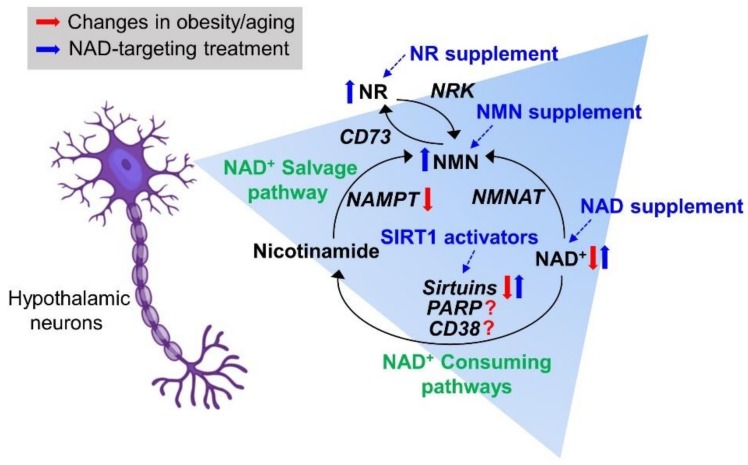
NAD^+^ biology-targeting treatment strategies for obesity- and aging-related metabolic and circadian disorders. CD38: cyclic ADP-ribose cyclase/cluster of differentiation 38, CD73: ecto-5′-nucleotidase/cluster of differentiation 73, NAD^+^: nicotinamide adenine dinucleotide, NAMPT, nicotinamide phosphoribosyltransferase, NMN, nicotinamide mononucleotide, NMNAT: nicotinamide mononucleotide adenylyltransferase, NR: nicotinamide riboside, NRK: nicotinamide riboside kinase, and PARP: poly(ADP-ribose) polymerases.

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
