# Peer review of "Hypothalamic NAD+-Sirtuin Axis: Function and Regulation"

_biomolecules, 2020, doi:10.3390/biom10030396_

Round 1

Reviewer 1 Report

I read this review with high enthusiasm. It is well structured and focused on the points the Authors wished to highlight and discuss. Indeed, NAD biology has been attracting the interest of many researchers and this review will be useful for many of them.

I just have very few points:

Line 59: in the case of CD38, NAD is “substrate”, and not co-substrate. Please specify.

Line 64: in this paragraph, Authors should also mention the fact that sirtuins are deacylating enzymes (in some cases, the deacylase activity is higher than the deacetylase activity; please refer, for instance, to Dr Denu studies)

Line 140: remove “a”

Author Response

Response to Reviewer 1.

Comments and Suggestions for Authors

I read this review with high enthusiasm. It is well structured and focused on the points the Authors wished to highlight and discuss. Indeed, NAD biology has been attracting the interest of many researchers and this review will be useful for many of them.

I just have very few points:

Line 59: in the case of CD38, NAD is “substrate”, and not co-substrate. Please specify.

Response: We revised the sentence as follows: “The activities of these enzymes are dependent on the cellular NAD+ levels as NAD+ is a substrate (CD38) or cosubstrate (sirtuins, PARPs) for the reactions of these enzymes.”

Line 64: in this paragraph, Authors should also mention the fact that sirtuins are deacylating enzymes (in some cases, the deacylase activity is higher than the deacetylase activity; please refer, for instance, to Dr Denu studies)

Response: We added the study results on the deacylase activity of sirtuins in Section 3 as follows: “SIRT6 effectively deacylates long-chain fatty acids whilst it has little deacetylase activity.”

Line 140: remove “a”

Response: We removed “a”.

Reviewer 2 Report

Summary

The article by Roh E and Kim MS entitled “Hypothalamic NAD biology in health and disease” aimed to review papers on the regulatory roles of hypothalamic neuronal NAD+-SIRT1 axis. Based on this primary objective and the title on the article, the review is not comprehensive enough to satisfy this aim unless revisions will be made.

Broad comments

The article is entitled “Hypothalamic NAD biology in health and disease” however the bulk of the text deals primarily on sirtuins and NAD+-consuming processes.  I suggest to change the title of the article to embody the general thought of the article.

I suggest to change the abbreviation for NAD to NAD+. Nicotinamide adenine dinucleotide exists either in reduced or oxidized forms. The oxidized form is the one essential for variety of processes being reviewed in the article.

The abstract stated that it is a comprehensive review however, the authors failed to add recent articles that will make it stand out from other review articles. First, the recent human trials on the potential use of NR for obesity (Dollerup OL et al, 2018, 2019, and 2020). It would be interesting to include these papers and compare the results to other human trials that the authors have mentioned in the article. And second, considering that the review article focus on NAD+ in hypothalamus, the authors have missed to include the study published this year (https://doi.org/10.1111/apha.13437) highlighting the essential role of hypothalamic NAMPT and food intake. It would be interesting to compare this study to the one done by the authors where NAD+ was centrally administered. This will add interesting insights to the review.

I highly recommend proofreading the entire article. There are several grammatical and spelling mistakes. Furthermore, sentence construction must be edited to make a more coherent transition from one section to another.

The authors should also consider revising some of the references that were cited as some of them do not cover stated facts.

Specific comments

  • Spelling/grammatical mistakes: Lines 10, 17-18
  • Consider deleting or changing the cited reference: L55 ref. 9, L58 ref. 14, L110 ref 54 & 56, L115 ref. 62,
  • Section 5: majority of what is reviewed here are not dealing with the NAD+ in the hypothalamus. I suggest editing the heading of the section
  • change the all the abbreviation for NAD to NAD+
  • I suggest editing the figures, increasing the font size of the texts especially for figure 1

Author Response

Reviewer 2.

Comments and Suggestions for Authors

Summary

The article by Roh E and Kim MS entitled “Hypothalamic NAD biology in health and disease” aimed to review papers on the regulatory roles of hypothalamic neuronal NAD+-SIRT1 axis. Based on this primary objective and the title on the article, the review is not comprehensive enough to satisfy this aim unless revisions will be made.

Broad comments

The article is entitled “Hypothalamic NAD biology in health and disease” however the bulk of the text deals primarily on sirtuins and NAD+-consuming processes. I suggest to change the title of the article to embody the general thought of the article.

Response: We will discuss with the editor about the title of article.

I suggest to change the abbreviation for NAD to NAD+. Nicotinamide adenine dinucleotide exists either in reduced or oxidized forms. The oxidized form is the one essential for variety of processes being reviewed in the article.

Response: We changed from NAD to NAD+.

The abstract stated that it is a comprehensive review however, the authors failed to add recent articles that will make it stand out from other review articles. First, the recent human trials on the potential use of NR for obesity (Dollerup OL et al, 2018, 2019, and 2020). It would be interesting to include these papers and compare the results to other human trials that the authors have mentioned in the article. And second, considering that the review article focus on NAD+ in hypothalamus, the authors have missed to include the study published this year (https://doi.org/10.1111/apha.13437) highlighting the essential role of hypothalamic NAMPT and food intake. It would be interesting to compare this study to the one done by the authors where NAD+ was centrally administered. This will add interesting insights to the review.

Response: Thank you for your helpful comments. We added the recent human study of NR supplement in obese, insulin resistant men in Section 7 as follows: “Contrary to these findings, 12 weeks-dietary NR supplementation did not alter insulin sensitivity, whole-body glucose metabolism, β-cell insulin secretory capacity and gut incretin hormone secretion in obese men with insulin-resistance [140,141]. Moreover, NR supplementation had no significant effect on skeletal muscle mitochondrial respiration, content or morphology in those subjects [142].”

We also added the recent study finding considering food intake regulation by hypothalamic NAMPT in Section 5 as follows: “According to a recent study [83], the hypothalamic expression levels of the NAD+ biosynthetic enzyme NAMPT are affected by administration of metabolic hormones (ghrelin, leptin) or HFD consumption. Furthermore, hypothalamic NAMPT inhibition with its chemical inhibitor FK866 abrogates nocturnal feeding and ghrelin-induced hyperphagia. These data suggest that the NAD+ salvage pathway in hypothalamic neurons may be involved in the adaptive regulation of feeding behavior to metabolic alterations.”

We compared this study results with ours in Section 7 as follows: “Our findings seem to be contradictory to the findings of a recent study [83]. In that study, intracerebroventricular administration of NAMPT inhibitor FK866 inhibited ghrelin-induced feeding and also suppressed nocturnal feeding when administered before light-off [83]. These conflicting results might be caused by the differences in experimental settings or animals. It should be noted, however, that SIRT1 in POMC and SF1 neurons acts to resist weight gain whereas SIRT1 in NPY/AGRP neurons are required for fasting- and ghrelin-induced hyperphagia [80]. Therefore, therapeutic elevation of hypothalamic NAD+ contents might bring differential outcomes according to the metabolic state during experiments.”

I highly recommend proofreading the entire article. There are several grammatical and spelling mistakes. Furthermore, sentence construction must be edited to make a more coherent transition from one section to another.

Response: We got an English edition service from the Boston Biodit Co and attached the certification.

The authors should also consider revising some of the references that were cited as some of them do not cover stated facts.

Response: We deleted and added some of the references.

Specific comments

Spelling/grammatical mistakes: Lines 10, 17-18

Response: We corrected the grammatical mistakes.

Consider deleting or changing the cited reference: L55 ref. 9, L58 ref. 14, L110 ref 54 & 56, L115 ref. 62,

Response: We deleted and changed the reference according to your opinion.

Section 5: majority of what is reviewed here are not dealing with the NAD+ in the hypothalamus. I suggest editing the heading of the section

Response: We edited the heading of the section 5 to “Regulatory roles of hypothalamic NAD+-sirtuin axis in normal physiological conditions”.

the all the abbreviation for NAD to NAD+

Response: We changed all the abbreviation for NAD to NAD+.

I suggest editing the figures, increasing the font size of the texts especially for figure 1

Response: We increased the fond sizes in figure 1 and 2.

Reviewer 3 Report

This review by Roh and Kim is a well-written account on hypothalamic NAD biology and its role in health and disease. Overall, the review is comprehensive and the main focus is on pre-clinical studies performed in different rodent models. The reader is introduced to the pathways synthesizing and consuming NAD+, and there is a nice introduction to the described role of NAD+ and SIRT1 in specific hypothalamic neurons for regulation of metabolism. While the authors mostly described rodent studies, they do also include data from human studies in this review. One weakness, however, is that this discussion appears to be biased and not complete. This also applies to some parts of the review dealing with studies in rodents. Thus, the reader is left with the impression that the review lack nuance. This should be addressed in a revised version of the manuscript. The following points should be addressed:

  1. In line 42 and 56 the authors mention that NAD+ can be degraded to nicotinic acid (NA). It is not commonly accepted that SIRTs, PARPs or CD38 can degrade NAD+ to NA. If this is not a misunderstanding, please provide specific references to support this claim.
  2. In lines 202-204, the authors state that mice on a HFD have perturbed NAMPT-mediated NAD+ biosynthesis. Several studies show the opposite. Work from Penke et al. 2015 (PMID 26033245), Dall et al. (PMID 29408602), and Dall et al. (PMID 31320478) show that NAMPT and/or NAD+ levels are enhanced or unaffected by HFD-feeding. These studies should be included and discussed in the relevant context.
  3. In line 223 the authors should include a reference to the recent paper by de Guia et al. (PMID 31207144) showing no decrease in NAMPT protein abundance in adipose tissue with age in humans.
  4. In lines 270-271 the authors reference human clinical trials with nicotinamide riboside and nicotinamide mononucleotide. However, the authors completely miss the recent studies conducted by Dollerup et al. (PMIDs 29992272, 31390002, 31710095) showing no effects of NR on multiple clinically relevant outcomes in obese (BMI >30), insulin resistant, and middle-aged men. These papers should be included in the discussion.
  5. In line 273 the authors states that NR can reduce blood pressure and aortic stiffness. In that study (i.e., reference 125), there were no statistical differences in either of these parameters between treatment groups. Thus, the statement should be modified.
  6. In line 290, the authors discuss their own data on central NAD administration in the brain. They have shown that NAD suppresses fasting-induced hyperphagia. A recent study from de Guia et al. 2020 (PMID 31900990) show that FK866 administration into the 3rd ventricle of the brain in mice reduces fasting- and ghrelin-induced food intake. This paper should be discussed.
  7. In line 311 the authors states that knockout of Nampt in adipose tissue reduces circulating eNAMPT levels. This also correct, but only in female mice, not in male mice as also shown in the paper they reference (reference 131). This difference between genders should be high-lighted.

Author Response

Reviewer 3.

Comments and Suggestions for Authors

This review by Roh and Kim is a well-written account on hypothalamic NAD biology and its role in health and disease. Overall, the review is comprehensive and the main focus is on pre-clinical studies performed in different rodent models. The reader is introduced to the pathways synthesizing and consuming NAD+, and there is a nice introduction to the described role of NAD+ and SIRT1 in specific hypothalamic neurons for regulation of metabolism. While the authors mostly described rodent studies, they do also include data from human studies in this review. One weakness, however, is that this discussion appears to be biased and not complete. This also applies to some parts of the review dealing with studies in rodents. Thus, the reader is left with the impression that the review lack nuance. This should be addressed in a revised version of the manuscript. The following points should be addressed:

Response: Thank you for your valuable comments. We have added more human study results in detail. We also reviewed the latest published study findings and compared them with our study results.

In line 42 and 56 the authors mention that NAD+ can be degraded to nicotinic acid (NA). It is not commonly accepted that SIRTs, PARPs or CD38 can degrade NAD+ to NA. If this is not a misunderstanding, please provide specific references to support this claim.

Response: We changed the description of NAD+ salvage pathway to avoid misunderstanding in Section 2 as follows: “Another important NAD+-biosynthetic pathway is the so-called salvage pathway, via which most cellular NAD+ is synthesized in mammals. This pathway reuses nicotinamide or nicotinic acid produced by enzyme reactions that degrade NAD+. In the NAD+ salvage pathway of mammals, nicotinamide is directly converted to nicotinamide mononucleotide (NMN) by nicotinamide phosphoribosyl transferase (NAMPT),”

In lines 202-204, the authors state that mice on a HFD have perturbed NAMPT-mediated NAD+ biosynthesis. Several studies show the opposite. Work from Penke et al. 2015 (PMID 26033245), Dall et al. (PMID 29408602), and Dall et al. (PMID 31320478) show that NAMPT and/or NAD+ levels are enhanced or unaffected by HFD-feeding. These studies should be included and discussed in the relevant context.

Response: We added those study results and discussed with the discrepant results in Section 6 as follows: “However, conflicting results have been reported in regard to hypercaloric diet- or aging-induced changes in metabolic organ NAMPT and NAD+ levels. Hepatic NAD+ contents and NAMPT expression were shown to be elevated in mice on a HFD and suggested as a compensatory mechanism to protect against hepatic lipid accumulation [111]. In other study, hepatic NAD+ and NAMPT levels were not altered by consumption of a HFD [112]. These discrepant results from animal studies might be caused by HFD composition and consumption duration as well as by rodent strains used. Conflicting human data on the metabolic organ NAMPT expression have been also reported. Obese women with hepatic steatosis showed higher hepatic NAMPT mRNA expression compared to lean women [113]. In a recent paper, aged humans exhibited reduced NAMPT protein levels in skeletal muscle but no change in adipose tissue compared with those of young subjects [114].”

In line 223 the authors should include a reference to the recent paper by de Guia et al. (PMID 31207144) showing no decrease in NAMPT protein abundance in adipose tissue with age in humans.

Response: We added the study results in Section 6 as follows: “In a recent paper, aged humans exhibited reduced NAMPT protein levels in skeletal muscle but no change in adipose tissue compared with those of young subjects [115].”

In lines 270-271 the authors reference human clinical trials with nicotinamide riboside and nicotinamide mononucleotide. However, the authors completely miss the recent studies conducted by Dollerup et al. (PMIDs 29992272, 31390002, 31710095) showing no effects of NR on multiple clinically relevant outcomes in obese (BMI >30), insulin resistant, and middle-aged men. These papers should be included in the discussion.

Response: Thank you for your helpful comments. We added the recent human study results using NR in obese, insulin resistant men in Section 7 as follows: “Contrary to these findings, 12 weeks-dietary NR supplementation did not alter insulin sensitivity, whole-body glucose metabolism, β-cell insulin secretory capacity and gut incretin hormone secretion in obese men with insulin-resistance [140,141]. Moreover, NR supplementation had no significant effect on skeletal muscle mitochondrial respiration, content or morphology in those subjects [142].”

In line 273 the authors states that NR can reduce blood pressure and aortic stiffness. In that study (i.e., reference 125), there were no statistical differences in either of these parameters between treatment groups. Thus, the statement should be modified.

Response: We modified the statement in Section 7 as follows: “and induced beneficial vascular changes: the trend towards decreased blood pressure and aortic stiffness [139].”

In line 290, the authors discuss their own data on central NAD administration in the brain. They have shown that NAD suppresses fasting-induced hyperphagia. A recent study from de Guia et al. 2020 (PMID 31900990) show that FK866 administration into the 3rd ventricle of the brain in mice reduces fasting- and ghrelin-induced food intake. This paper should be discussed.

Response: We added the recent study finding from de Guia et al. in Section 5 as follows: “According to a recent study [83], the hypothalamic expression levels of the NAD+ biosynthetic enzyme NAMPT are affected by administration of metabolic hormones (ghrelin, leptin) or HFD consumption. Furthermore, hypothalamic NAMPT inhibition with its chemical inhibitor FK866 abrogates nocturnal feeding and ghrelin-induced hyperphagia. These data suggest that the NAD+ salvage pathway in hypothalamic neurons may be involved in the adaptive regulation of feeding behavior to metabolic alterations.”

We compared this study results with ours in Section 7 as follows: “Our findings seem to be contradictory to the findings of a recent study [83]. In that study, intracerebroventricular administration of NAMPT inhibitor FK866 inhibited ghrelin-induced feeding and also suppressed nocturnal feeding when administered before light-off [83]. These conflicting results might be caused by the differences in experimental settings or animals. It should be noted, however, that SIRT1 in POMC and SF1 neurons acts to resist weight gain whereas SIRT1 in NPY/AGRP neurons are required for fasting- and ghrelin-induced hyperphagia [80]. Therefore, therapeutic elevation of hypothalamic NAD+ contents might bring differential outcomes according to the metabolic state during experiments.”

In line 311 the authors states that knockout of Nampt in adipose tissue reduces circulating eNAMPT levels. This also correct, but only in female mice, not in male mice as also shown in the paper they reference (reference 131). This difference between genders should be high-lighted.

Response: We described the results of the study divided by gender as follows: “Interestingly, these changes were observed only in females and the mechanism of the gender difference remained unknown. Conversely, adipose tissue-specific NAMPT knockin mice showed opposite changes in these parameters, which were observed in both males and females.”

Round 2

Reviewer 3 Report

The authors have addressed by previous concerns in a satisfactory manner.